# Effect of Vedolizumab on Anemia of Chronic Disease in Patients with Inflammatory Bowel Diseases

**DOI:** 10.3390/jcm9072126

**Published:** 2020-07-06

**Authors:** Patrizio Scarozza, Elena De Cristofaro, Ludovica Scucchi, Irene Rocchetti, Irene Marafini, Benedetto Neri, Silvia Salvatori, Livia Biancone, Emma Calabrese, Giovanni Monteleone

**Affiliations:** 1Department of Systems Medicine, University of Rome “Tor Vergata”, 00133 Rome, Italy; scarozzapatrizio@gmail.com (P.S.); elena_decr@hotmail.it (E.D.C.); ludovicascucchi@yahoo.it (L.S.); irene.marafini@gmail.com (I.M.); benedettoneri@gmail.com (B.N.); silviasalvatori23@gmail.com (S.S.); biancone@med.uniroma2.it (L.B.); emma.calabrese@uniroma2.it (E.C.); 2Statistical Office, Superior Council of Judiciary, 00185 Rome, Italy; irene.rocchetti@gmail.com

**Keywords:** Crohn’s disease, ulcerative colitis, biologics, anemia

## Abstract

Background: Anemia of Chronic Disease (ACD) can negatively influence the clinical course of Inflammatory Bowel Disease (IBD) patients. The aim of this study was to evaluate the effect of Vedolizumab on ACD in IBD. Methods: Clinical data of 75 IBD patients (25 Crohn’s disease (CD) and 50 Ulcerative Colitis (UC)) receiving Vedolizumab in a tertiary referral IBD center were retrospectively evaluated and the effect of the drug on ACD was ascertained at weeks 14 and 24. Results: ACD was diagnosed in 35 (11 CD and 24 UC) out of 75 (47%) IBD patients. At both week 14 and week 24, improvements and resolutions of ACD were achieved by 13/35 (37%) and 11/35 (31%) patients, respectively. Baseline demographic/clinical characteristics did not differ between patients with ACD improvements/resolutions and those with persistent ACD. Clinical response occurred more frequently in patients who achieved ACD resolution (10/11, 91%) than in those without ACD improvement (5/11, 45%, *p* = 0.022). When analysis was restricted to anemic patients, ACD resolution was documented in 10/22 patients (45%) achieving clinical response and 1/13 of non-responders (8%; *p* = 0.02). Conclusions: ACD occurs in half of the IBD patients and, in nearly two thirds of them, Vedolizumab treatment associates with ACD resolution/improvement.

## 1. Introduction

Anemia of chronic disease (ACD) (also referred to as anemia of inflammation) is a form of anemia, which develops in the context of systemic inflammation because of reduced production of erythrocytes, accompanied by a modest reduction in erythrocyte survival [1]. In ACD, the erythrocytes are generally normal and not small (low mean corpuscular volume) and hemoglobin-deficient (low mean corpuscular hemoglobin concentration), as seen in iron-deficiency anemia (IDA). However, erythrocytes can become small in the cases of ACD, in which iron deficiency coexists or develops as a complication. Similar to IDA, ACD is characterized by low serum iron levels, but it differs from IDA in that iron stores are preserved in macrophages [1]. Therefore, ACD is primarily a disorder of iron distribution and cellular metabolism.

Iron metabolism and homeostasis are tightly controlled phenomena, which are mainly under the control of hepcidin, produced by hepatocytes, and ferroportin, which is both the hepcidin receptor and the sole cellular iron exporter expressed on the cell surface of macrophages, hepatocytes and enterocytes, through which iron is transferred from the intracellular compartment to the blood [2,3,4]. Hepcidin inhibits the activation of ferroportin, thereby promoting the accumulation of iron in iron-recycling macrophages. During chronic inflammation, cytokines, such as interleukin (IL)-6 and tumor necrosis factor (TNF), enhance production of hepcidin, with the downstream effect of limiting the availability of iron for bone marrow erythropoiesis [1,5]. TNF can also convert erythropoiesis to myelopoiesis in human hematopoietic stem/progenitor cells, further contributing to ACD [6]. Moreover, chronic inflammatory processes are associated with the diminished renal production of erythropoietin and a decreased expression of erythropoietin receptors on erythroid progenitors [7]. The hyperactivation of macrophages by inflammatory stimuli induces hemophagocytosis and, consequently, a diminished erythrocyte lifespan [1].

Both ACD and IDA are common systemic complications of inflammatory bowel diseases (IBD) and have been associated with restless leg syndrome, fatigue, impaired physical function, decreased quality of life (QoL) and cognitive function in IBD patients [2]. Indeed, therapeutic interventions aimed at increasing hemoglobin resulted in improved QoL scores, independent of IBD activity [8]. On the other hand, it has been shown that the prevalence and severity of anemia are related to IBD activity and treatments used to attenuate the IBD-associated mucosal inflammation (i.e., TNF blockers) can improve ACD [9,10,11].

Vedolizumab, a gut-selective humanized monoclonal antibody that binds to the α4β7 integrin and selectively reduces intestinal immune cell trafficking, is a safe and effective treatment option for patients with IBD [12,13,14,15,16,17]. In both ulcerative colitis (UC) and Crohn’s disease (CD), Vedolizumab is effective in inducing and maintaining clinical and endoscopic/histologic remission [12,13,14,15,16,17]. However, to the best of our knowledge, no study has yet evaluated the effect of Vedolizumab on ACD, although, at least in CD, anemia at baseline has been associated with lower durability of treatment [18]. We here examine the effect of Vedolizumab on ACD.

## 2. Materials and Methods

### 2.1. Study Design

This was a retrospective study conducted on IBD patients treated with Vedolizumab at the Tor Vergata University Hospital (Rome, Italy). Patients’ data were retrospectively collected between April 2018 and October 2019 and, after a de-identification process, registered into an electronic database. The primary objective of the study was to evaluate the effect of Vedolizumab on ACD improvement and resolution at weeks 14 and 24 of therapy. The week 24 was selected because the data came from clinical charts, which were completed between infusion sessions at weeks 22 and 30. The study was approved by the local Ethics Committee (CEI Policlinico Tor Vergata, Rome) (code 0024988/2019; 14 January 2020).

### 2.2. Patients

Inclusion criteria included: a confirmed diagnosis of CD or UC [19,20]; a clinically active disease at baseline (regardless of the grade) requiring Vedolizumab treatment; available data on clinical outcome at baseline and at weeks 14 and 24 of therapy. Patients were excluded if they were in clinical remission at baseline, had unclassified/indeterminate colitis or pouchitis and if the clinical data at the indicated time points were not available.

For each patient, several demographic and clinical variables were considered for the analysis, as shown in Appendix A. Clinical disease activity for UC was evaluated by the partial Mayo (pMayo) score (mild activity: pMayo of 2–4; moderate activity: pMayo of 5–7; severe activity: pMayo > 7) [21] and for CD by the Harvey–Bradshaw index (HBI) (mild activity: HBI of 5–7; moderate activity: HBI of 8–16; severe activity: HBI > 16) [22]. Clinical response was defined as a reduction of a minimum of three points of the pMayo score for UC and HBI for CD. Endoscopic activity at baseline was evaluated by the endoscopic Mayo score for UC [23] and the Simple Endoscopic Score for CD (SES-CD) (mild activity: SES-CD score of 3–6; moderate activity: SES-CD of 7–15; severe activity: SES-CD > 15) [24].

ACD was defined as the presence of clinical evidence of inflammation with a hemoglobin level <13 gr/dL (for males) and <12 gr/dL (for females) and a serum ferritin >100 μg/L and transferrin saturation (TfS) <20% [2]. Mixed type anemia was defined as the presence of the abovementioned criteria associated with a serum ferritin level between 30 and 100 μg/L [2]. ACD improvement was defined as the increase in hemoglobin level by at least 1 gr/dL. ACD resolution was defined as the achievement of a normal value of hemoglobin (≥12 gr/dL for females and ≥13 gr/dL for males).

### 2.3. Statistical Analysis

Continuous variables were reported as median with interquartile range (IQR) and categorical variables were expressed as percentage. The patients without anemia improvement or resolution were considered as the group of comparison; the distribution of the variables between patients with ACD improvement or resolution (considered separately) and patients with persistent ACD at week 14 and week 24 were evaluated by binomial analysis, using the χ^2^ or Fisher exact test for the categorical variables and with Mann–Whitney test for the continuous variables. A *p* < 0.05 level was considered for statistical significance.

## 3. Results

### 3.1. Frequency of ACD in IBD

Seventy-five IBD patients (25 CD and 50 UC) were enrolled. Patients had a median duration of disease longer than 10 years and most of them (68%) had been previously exposed to TNF-α antagonists, as shown in Appendix A.

ACD was diagnosed in 35/75 (47%) patients (11 CD and 24 UC). Fifteen out of 35 patients (43%) had pure ACD, while the remaining 20 (57%) had mixed type anemia (ACD combined with IDA). Among the anemic patients, anemia was mild (≥9.5 gr/dl) in 31 patients (88%) and moderate (8–9.5 gr/dL) in the remaining patients (12%); no cases of severe anemia (<8 gr/dl) were recorded.

Demographic and clinical characteristics at baseline did not differ between patients with ACD and those without ACD as well as between patients with pure ACD and those with mixed type anemia, as shown in Table 1 and Appendix A, except for a higher level of both ferritin and transferrin saturation in the group of patients without ACD, as shown in Table 1. Concomitant immune–inflammatory disorders did not differ between patients with ACD and those without ACD, as shown in Table 1.

### 3.2. Effect of Vedolizumab on ACD Course

The clinical response to Vedolizumab was documented in 44/75 (59%; 11 CD and 33 UC) patients at week 14 and in 43/75 (57%) (11 CD and 32 UC) patients at week 24. At week 14, clinical response was observed in 22/35 (63%) patients with ACD and 22/40 (55%) of those without ACD, as shown in Table 1. At week 24, the percentage of responders did not differ between patients with ACD (22/35 (63%)) and those without ACD (21/40 (52%)), as shown in Table 1.

ACD improvement occurred in 13/35 (37%) patients at week 14 and was maintained in all of them at week 24. ACD improvement was documented in seven out of 15 patients with pure ACD (47%) and six out of 20 patients with mixed type anemia (30%) (*p* = 0.312), as shown in Appendix A. Patients with no improvement of ACD at week 14 remained anemic at week 24. There was no difference in terms of demographic and clinical characteristics between patients with and without ACD improvement, except for a higher CRP value at baseline in the group of patients with ACD improvement, as shown in Table 2.

ACD resolution was achieved by 11/35 (31%) patients at both week 14 and week 24. Four out of 15 patients with pure ACD (27%) and seven out of 20 patients with mixed type anemia (35%) achieved a resolution of anemia (*p* = 0.599), as shown in Appendix A. The baseline clinical and demographic characteristics did not differ between patients with ACD resolution and those with persistence of ACD, following Vedolizumab treatment, as shown in Table 3.

In line with the above results, the median values of hemoglobin increased following Vedolizumab treatment, even though a statistically significant difference was seen between baseline (median value: 10.9; interquartile range: 10.3–12.7) and week 14 (median value: 12; interquartile range: 11–13.5; *p* = 0.016) but not week 24 (median value: 11.9; interquartile range: 11.2–13.8; *p* = 0.186), as shown in Figure 1.

### 3.3. Relationship between Clinical Response and ACD Resolution

At both week 14 and week 24, clinical response occurred more frequently in patients who achieved ACD resolution (10/11, 91%) than in those without anemia improvement (5/11, 45%, *p* = 0.022) (Table 3). When analysis was restricted to the 35 anemic patients, ACD resolution was documented in 10 out of the 22 patients (45%) achieving clinical response and 1 out of the 13 non-responders (8%; *p* = 0.02). Among the 22 patients with clinical response to the drug there was no difference in the frequency of concomitant immuno-inflammatory disorders between patients achieving anemia improvement/resolution and those without anemia improvement (2/17 (12%) vs 1/5 (20%), *p* = 0.637).

## 4. Discussion

ACD is a condition that can accompany immune–inflammatory diseases, in which there is a decrease in hemoglobin, hematocrit and erythrocyte counts due to a complex process, usually initiated by cellular immunity mechanisms and inflammatory cytokines. Biologics used to treat such disorders are supposed to improve ACD due to their systemic activity [1,9]. This study was undertaken to evaluate the impact of Vedolizumab therapy on the course of ACD in IBD. Indeed, it is well known that ACD can complicate the course of both CD and UC and circumstantial evidence indicates that the development of ACD is mainly related to the IBD-associated inflammation [2]. We found a 47% prevalence of ACD. Most patients had mild anemia and this relies probably on the fact that the study was focused on IBD outpatients, thus excluding the most severe cases of ACD requiring hospitalization. ACD was more frequent in UC than in CD, consistent with the demonstration that circulating levels of hepcidin in UC patients are similar or higher than those in Crohn’s disease [25,26].

As with IBD, other chronic immuno-inflammatory disorders can be associated with ACD [27]. However, the frequency of concomitant immuno-inflammatory pathologies did not differ between patients with ACD and those without anemia.

The induction of clinical response at week 14 was achieved by half of the patients treated with Vedolizumab, thus confirming previous real-life study findings [28,29,30,31,32,33,34,35,36,37,38,39,40,41]. A substantial positive effect of Vedolizumab on ACD was seen in two thirds of the anemic patients. The resolution or improvement of ACD were observed at week 14 and maintained at week 24. The baseline clinical and demographic characteristics of the patients did not influence the effect of Vedolizumab on ACD, even though we cannot exclude the possibility that some associations could be masked by the small number of patients analyzed.

Previous studies have evaluated the therapeutic effect of TNF blockers on hemoglobin levels and anemia in IBD patients. In a study on 18 CD patients with anemia, treatment with infliximab improved hemoglobin levels in nearly two thirds of the cases [9]. Similarly, Lönnkvist and colleagues reported that CD responders to infliximab exhibited increased hemoglobin levels [42]. In contrast, no significant difference in hemoglobin levels was found between responders and non-responders to anti-TNF treatment (infliximab or adalimumab) in pediatric IBD [43]. In line with the latter results, Koutroubakis and colleagues showed that anti-TNF therapy had only a modest effect on patients’ hemoglobin levels, despite significant beneficial effects on disease activity and clinical outcomes [44]. The factors accounting for such discrepancies remain unknown, but differences in the patient selection and types of anemia considered might have contributed. In contrast to other chronic inflammatory disorders where anemia results mainly from the action of inflammatory molecules on hemopoiesis, anemia in IBD also relies on mucosal blood loss and impaired iron absorption. Our analysis was restricted to ACD, even though in more than 50% of the ACD patients there was a concomitant iron deficiency anemia. Nonetheless, the data indicate that ACD improvement and resolution following Vedolizumab treatment occurred independently of the baseline hemoglobin value and concomitant iron supplementation therapy, suggesting that the positive effect of the drug on ACD is secondary to the control of the ongoing mucosal inflammation. Indeed, we found a strong association between ACD resolution and clinical response to the drug, as 10/11 patients with ACD resolution achieved clinical response following Vedolizumab treatment. However, at both week 14 and week 24, in nearly half of the patients achieving clinical response, we documented no effect of the drug on ACD. Although it remains to be clarified, it is plausible that the persistence of ACD in such patients relies on the inability of the drug to fully halt the mucosal inflammation. It is also unlikely that the persistence of anemia in patients with clinical response was due to a concomitant immuno-inflammatory disorder because such pathologies were equally distributed between patients with ACD resolution/improvement and those with no ACD improvement.

This study has some limitations. It was a retrospective study based on data collected from the medical records of a small group of IBD patients and, therefore, we cannot exclude the fact that it was subjected to some selection biases, which could have either overestimated or underestimated the relationship between Vedolizumab treatment and ACD improvement. The effectiveness of Vedolizumab treatment was determined using only clinical scores as data on endoscopic/histological response to the treatment, and changes in inflammatory markers, such as serum CRP and fecal calprotectin, were not available. Therefore, it remains possible that some of the positive effects of the treatment on the clinical symptoms were not paralleled by a concomitant suppression of the ongoing mucosal inflammation, thereby introducing a bias in the definition of response/remission to Vedolizumab. Moreover, we had no blood samples to measure the circulating levels of hepcidin and to assess whether the effect of Vedolizumab on ACD was paralleled by the decreased synthesis of this protein. The fact that virtually all the patients had mild anemia may have also led to an overestimation of the effectiveness of the drug in ACD. Further limitations would include the relatively medium-term outcomes (i.e., 24 weeks) of the study and the lack of data on the relationships between the resolution of ACD and improvement in the quality of life of the patients, as it was proven that ACD significantly worsens the quality of life of patients with chronic diseases [2]. Therefore, we are aware that the present data may be used as the initial study generating hypotheses to be studied further by larger prospective studies.

In conclusion, this is the first study suggesting a positive effect of Vedolizumab on ACD course.

## Figures and Tables

**Figure 1 jcm-09-02126-f001:**
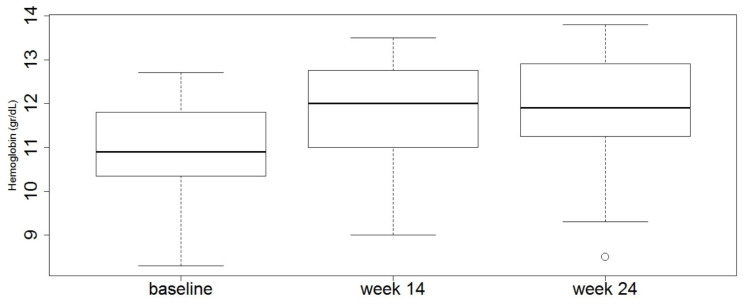
Vedolizumab treatment enhances hemoglobin values in patients with inflammatory bowel disease. The box-plots show the median values and the interquartile ranges at baseline and at week 14 and week 24 following Vedolizumab treatment in the total ACD population. Baseline vs. week 14, *p* = 0.016; baseline vs. week 24, *p* = 0.18.

**Table 1 jcm-09-02126-t001:** Distribution of baseline demographic/clinical characteristics and clinical response to Vedolizumab in patients with ACD and those without ACD.

	Patients with ACD (35/75)	Patients without ACD (40/75)	*p* Value
Male gender	15 (43%)	21 (52%)	*p* = 0.695
Age < 65 years	31 (88%)	31 (77%)	*p* = 0.206
Crohn′s disease	11 (31%)	14 (35%)	*p* = 0.743
Ulcerative colitis	24 (69%)	26 (65%)	*p* = 0.743
Current smokers	3 (8%)	6 (15%)	*p* = 0.392
Previous anti-TNF	26 (74%)	25 (62%)	*p* = 0.275
Concomitant steroids	18 (51%)	23 (57%)	*p* = 0.598
Concomitant immunosuppressors	1 (3%)	5 (12%)	*p* = 0.124
Concomitant immuno-inflammatory disorders ^§^	5 (14%)	4 (10%)	*p* = 0.324
Severe clinical activity	3 (8%)	1 (2%)	*p* = 0.243
Moderate clinical activity	23 (66%)	29 (73%)	*p* = 0.524
Mild clinical activity	9 (26%)	10 (25%)	*p* = 0.943
Severe endoscopic activity *	21 (66%)	23 (66%)	*p* = 0.993
Moderate endoscopic activity	3 (9%)	9 (26%)	*p* = 0.081
Mild endoscopic activity	8 (25%)	3 (8%)	*p* = 0.069
Hemoglobin (median, IQR) (gr/dL)	10.9 (10.35–12.7)	13.8 (13.2–15.1)	*p* = 0.0002
Ferritin value (median, IQR) (µg/L)	86 (45–143)	103 (84.7–189)	*p* = 0.019
Transferrin saturation (median, IQR) (%)	16 (13.2–20)	22 (21.6–37.1)	*p* = 0.002
CRP > 5 mg/L	23 (66%)	20 (50%)	*p* = 0.169
CRP value (median, IQR) (mg/L)	10 (3.75–56.3)	6.8 (2.85–52)	*p* = 0.849
IBD clinical response to Vedolizumab at week 14	22 (63%)	22 (55%)	*p* = 0.490
IBD clinical response to Vedolizumab at week 24	22 (63%)	21 (52%)	*p* = 0.365

ACD: Anemia of Chronic Disease; Anti-TNF: Anti-Tumor Necrosis Factor; CRP: C reactive protein; IBD: Inflammatory Bowel Disease; ^§^ Concomitant immuno-inflammatory disorders included three Hashimoto thyroiditis, one autoimmune pancreatitis and one erythema nodosum in the group of ACD and one rheumatoid arthritis, one Basedow disease, one ankylosing spondylitis and one Hashimoto thyroiditis in the group of patients without anemia; * Endoscopic data available in 32/35 patients with ACD and in 35/40 patients without ACD; IQR: interquartile range.

**Table 2 jcm-09-02126-t002:** Distribution of baseline demographic/clinical characteristics and clinical response to Vedolizumab in patients with ACD improvement and those without ACD improvement.

Variable	Patients with ACD Improvement (13/35)	Patients without ACD Improvement (11/35)	*p* Value
Male gender	7 (54%)	4 (36%)	*p* = 0.391
Age < 65 years	10 (77%)	11 (100%)	*p* = 0.222
Crohn′s Disease	7 (54%)	2 (18%)	*p* = 0.072
Ulcerative colitis	6 (46%)	9 (82%)	*p* = 0.072
Current smokers	1 (8%)	0	*p* = 1
Previous anti-TNF	11 (85%)	7 (64%)	*p* = 0.236
Concomitant steroids	8 (61%)	5 (45%)	*p* = 0.430
Concomitant immunosuppressors	1 (8%)	0	*p* = 1
Hemoglobin (median, IQR) (gr/dL)	10.9 (8.7–12.7)	11.4 (8.9–12.3)	*p* = 0.865
Mild anemia (hemoglobin ≥ 9.5 gr/dL)	11 (85%)	10 (91%)	*p* = 0.642
Moderate anemia (hemoglobin 8–9.5 gr/dL)	2 (15%)	1 (9%)	*p* = 0.642
Severe clinical activity ^∫^	1 (8%)	2 (18%)	*p* = 0.438
Moderate clinical activity	10 (77%)	5 (45%)	*p* = 0.112
Mild clinical activity	2 (15%)	4 (36%)	*p* = 0.236
Severe endoscopic activity *	7 (58%)	9 (82%)	*p* = 0.221
Moderate endoscopic activity	5 (42%)	1 (9%)	*p* = 0.075
Mild endoscopic activity	0	1 (9%)	*p* = 0.478
CRP > 5 mg/L	10 (77%)	5 (45%)	*p* = 0.112
CRP value (median, IQR) (mg/L)	10.5 (6–28)	3.9 (1.8–40)	*p* = 0.033
Iron therapy for anemia	6 (46%)	7 (64%)	*p* = 0.391
IBD clinical response to Vedolizumab at week 14	7 (54%)	5 (45%)	*p* = 0.682
IBD clinical response to Vedolizumab at week 24	7 (54%)	5 (45%)	*p* = 0.682

ACD: Anemia of Chronic Disease; Anti-TNF: Anti-Tumor Necrosis Factor. CRP; C reactive protein; IBD: Inflammatory Bowel Disease; IQR: Interquartile range; ^∫^ Clinical activity was classified with partial Mayo Score for ulcerative colitis (mild activity: pMayo of 2–4; moderate activity: pMayo of 5–7; severe activity: pMayo > 7) and with Harvey–Bradshaw Index (HBI) for Crohn’s disease (mild activity: HBI of 5–7; moderate activity: HBI of 8–16; severe activity: HBI > 16). * Endoscopic data classified with Endoscopic Mayo Score for ulcerative colitis and Simple Endoscopic Score for Crohn’s disease (SES-CD) available in 12/13 patients with ACD improvement and in 11/11 patients without ACD improvement. A SES-CD score of 3–6 was considered as mild endoscopic activity, 7–15 as moderate endoscopic activity and > 15 as severe endoscopic activity.

**Table 3 jcm-09-02126-t003:** Distribution of baseline demographic/clinical characteristics and clinical responses to Vedolizumab in patients with ACD resolution, as compared to patients without ACD improvement.

Variable	Patients with ACD Resolution (11/35)	Patients without ACD Improvement (11/35)	*p* Value
Male gender	4 (36%)	4 (36%)	*p* = 1
Age < 65 years	10 (91%)	11 (100%)	*p* = 1
Crohn′s Disease	2 (18%)	2 (18%)	*p* = 1
Ulcerative colitis	9 (82%)	9 (82%)	*p* = 1
Current smokers	1 (9%)	2 (18%)	*p* = 0.534
Previous anti-TNF	8 (73%)	7 (64%)	*p* = 0.647
Concomitant steroids	5 (45%)	5 (45%)	*p* = 1
Concomitant immunosuppressors	0	0	*p* = 1
Hemoglobin (median, IQR) (gr/dL)	10.9 (8.3–12.5)	11.4 (8.9–12.3)	*p* = 0.373
Mild anemia (hemoglobin ≥ 9.5 gr/dL)	10 (91%)	10 (91%)	*p* = 1
Moderate anemia (hemoglobin 8–9.5 gr/dL)	1 (9%)	1 (9%)	*p* = 1
Severe clinical activity ^∫^	0	2 (18%)	*p* = 0.476
Moderate clinical activity	8 (73%)	5 (45%)	*p* = 0.193
Mild clinical activity	3 (27%)	4 (36%)	*p* = 0.647
Severe endoscopic activity *	5 (56%)	9 (82%)	*p* = 0.202
Moderate endoscopic activity	2 (22%)	1 (9%)	*p* = 0.413
Mild endoscopic activity	2 (22%)	1 (9%)	*p* = 0.413
CRP > 5 mg/L	8 (73%)	5 (45%)	*p* = 0.193
CRP value (median, IQR) (mg/L)	13.3 (5.65–56.3)	3.9 (1.8–40)	*p* = 0.138
Iron therapy for anemia	4 (36%)	7 (64%)	*p* = 0.200
IBD clinical response to Vedolizumab at week 14	10 (91%)	5 (45%)	*p* = 0.022
IBD clinical response to Vedolizumab at week 24	10 (91%)	5 (45%)	*p* = 0.022

ACD: Anemia of Chronic Disease; Anti-TNF: Anti-Tumor Necrosis Factor; CRP: C reactive protein; IBD: Inflammatory Bowel Disease; IQR: Interquartile range. ^∫^ Clinical activity was classified with partial Mayo Score for ulcerative colitis (mild activity: pMayo of 2–4; moderate activity: pMayo of 5–7; severe activity: pMayo > 7) and with Harvey–Bradshaw Index (HBI) for Crohn’s disease (mild activity: HBI of 5–7; moderate activity: HBI of 8–16; severe activity: HBI > 16). * Endoscopic data (classified with Endoscopic Mayo Score for ulcerative colitis and Simple Endoscopic Score for Crohn’s disease) available in 9/11 patients with ACD resolution and in 11/11 patients without ACD improvement. A SES-CD score of 3–6 was considered as mild endoscopic activity, 7–15 as moderate endoscopic activity and > 15 as severe endoscopic activity.

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
