# Peer review of "Effect of Vedolizumab on Anemia of Chronic Disease in Patients with Inflammatory Bowel Diseases"

_jcm, 2020, doi:10.3390/jcm9072126_

Round 1
Reviewer 1 Report
Authors discussed a topic which is not a very “hot topic” in IBD. Indeed, as ACD is referred as anemia of inflammation, it makes sense that biologics targeting inflammatory pathways will positively act on ACD! However, we know that observation is the key in science. So, I was interesting to read how authors will develop this topic and I have been quite disappointed.
The text is not easy to read with too many tables (not instructive), an absence of graphs and figures (the only figure is without any interest), an absence of correlation with inflammatory markers (it is a pity when studying ACD…).
Also, the study is retrospective with few ACD patients. It is not clear how many data are missing
Major comments
- The authors well introduced the ACD with recent nice review. This is an anemia referred as anemia of inflammation. But authors evaluated the efficacy of vedolizumab on ADC by analyzing the clinical response only (based on imperfect surrogates such as HBI or pMayo score). To establish a valid correlation between ACD resolution and vedolizumab efficacy, a criteria including both clinical response AND inflammatory markers should be used.
- Quid evolution of inflammatory markers (such as CRP or CF) to have an objective marker related to correction of ACD? Nothing is clearly represented trough the paper although it is closely related to ACD resolution
- Results 3.1 : 35 patients of ACD including 15 pure and 20 mixed. None patient with anemia related to iron deficiency without criteria for ACD? It is quite difficult to imagine. When authors said “among anemic patients”, that is related to 35 ACD patients? No other patient with anemia from another cause?
- It is interesting to see that more UC patients have ACD than CD patients. In a first view, we could imagine that more UC patients have pure “iron deficiency anemia” knowing that (1) systemic inflammation is often less pronounced in UC patients and that (2) bloody stool are more frequent. A table comparing the two diseases with ACD should be interesting with description of CRP, CF if available, endoscopic activity. And this point should be discussed to explain which systemic cytokines/interleukines or mechnisms could help to explain this observation
- Table 1 has to be improved:
Repartition of UC to add in table 1 (as well as in table S2)
The item “increased CRP value” should be quantified to ease the comparison between the two groups
The biologic criteria for ACD should be presented in the table to have a comparison between ACD and not ACD (level of Hb, serum ferritin, transferrin saturation). Also, biologic value for inflammation should be added (CRP level, CF, platelets, leucocytes)
- Figure 1 has few interests due to the constancy of numbers between week 14 and 24. The authors should replace this figure by a graph showing the evolution (at baseline, week 14 and week 24) of Hb, CRP and CF in the different groups
- Between line 153 and 155, the numbers exposed by authors are not easy to find in the different tables. As this information seem pertinent for the authors, the numbers should be more highlighted in tables.
- In paragraph “Effect of Vedolizumab on ACD Course”, it appears that all patients maintained vedolizumab up to week 24. In the opposite, In paragraph “Relationship between Clinical Response and ACD Resolution”, the lector learns that 16 patients were non responders, what does it mean?
- The discrepancy between ADC improvement and ADC resolution is confusing. Probably it is related to a power issue. A graphic representation of haemoglobin resolution with for instance box and whisker plot could help to understand this absence of significance.
- In the Discussion part, authors summarise some results which are hard to find in the results section:
Line 165-166: “a A substantial positive effect of Vedolizumab on ACD was seen in two thirds of the anemic patients.” Really, two-third? It does not appear clearly the results…
Line 167- 168 “Interestingly, resolution or improvement of ACD were observed at week 14 and maintained at week 24, and the patients who did not benefit from Vedolizumab after the induction therapy remained anemic at week 24. “ Authors cannot conclude this knowing that there is no difference between the groups at improvement level
- In the discussion part, a critical view of physiopathology of ACD should be discussed. Indeed, it seems quite logical that drugs such as infliximab and vedolizumab improve ACD because these biologics have been validated to improve systemic inflammation in IBD
- It could be more constructive to reminder this point at the beginning and then discuss why this relation is not always observed (due to methodology, primary non response, underlined other cause of anemia,…)”
Minor comments
- The classification for disease activity should be defined by using the ranges of mayo score and SES-CD in methods section.
- Table 2 and 3 could be merged knowing that the column patients w/o ACD is common in both tables
Reviewer 2 Report
Thank you for the invitation to review this manuscript by Scarozza and colleagues. The study is well conceived, presented in a rational manner and addresses a relatively novel aspect of IBD management. The introduction section is also particularly helpful, as the molecular mechanisms underlying anaemia of chronic disease may not be familiar to many gastroenterologists.
I think the overall conclusion of the study, that vedolizumab is associated with resolution of ACD, is interesting and may be of use in some specific scenarios but is unlikely to impact greatly on clinical decision-making in general. However, as the authors point out, this finding has not been previously reported and so, forms a novel addition to the literature.
I have a couple of queries/suggestions that may help clarify the study report:
- Did any of the patients have iron infusions during the study period? Were any of them taking oral iron supplements? If not, were these exclusion criteria?
- I find the timepoints slightly difficult to understand - week 14 makes sense as patients attend for infusions then, but why was week 24 chosen? Surely that would be between infusions given at weeks 22 and 30. This should be clarified.
- I commend use of Mayo and in particular SES-CD to assess endoscopic activity but I don't believe we're given definitions for the mild, moderate and severe stratifications in table 2. Mayo is commonplace enough not to necessarily need definitions (although I still think these should be included) but SES-CD probably does.
- Similarly, stratifications for clinical disease activity (using HBI or pMayo) are given in table 2 but not defined. They should be.
- The statement "ACD was defined as the presence of clinical evidence of inflammation with a Hb..." needs further definition as per point 4. Was the clinical evidence of inflammation a HBI/pMayo above a certain threshold? If so, what threshold(s)?
- Were co-morbidities collected? Surely these impact on ACD - particularly inflammatory co-morbidities (e.g. RA) that would be expected to be more common in this population and would not necessarily be expected to respond to vedolizumab's mechanism of action. This would be an interesting addition to Table 1.
- The limitations should include the relatively short- (or, at best medium-) term outcomes of the study and lack of quality of life assessments.
- The conclusion of the abstract should probably be softened to say that Vedolizumab is "associated with" ACD resolution/improvement, rather than "promotes".
Round 2
Reviewer 1 Report
The authors made significant changes and the paper is clearer with more nuances. Figure 1 is more appropriate than the previous one and discussion has been improved.
Thank you to authors for their efforts.
I have no more comments
Reviewer 2 Report
Thank you for making the suggested revisions. I believe they have improved the manuscript significantly.